# Dark Fermentation in the Dark Biosphere: The Case of *Citrobacter* sp. T1.2D-1[2]

**Violeta Gallego-Rodríguez [1,†], Adrián Martínez-Bonilla [1,†], Nuria Rodríguez [1,2] and Ricardo Amils [1,2,*]**

[1]  Centro de Biología Molecular Severo Ochoa (CSIC-UAM), Universidad Autónoma de Madrid, 28049 Madrid, Spain; violetagallego6@gmail.com (V.G.-R.); amartinez@cbm.csic.es (A.M.-B.); nrodriguez@cbm.uam.es (N.R.)
[2]  Centro de Astrobiología (CAB), CSIC-INTA, 28850 Torrejón de Ardoz, Spain
[*]  Correspondence: ramils@cbm.csic.es
[†]  These authors contributed equally to this work.

**Abstract:** Microbial diversity that thrives in the deep subsurface remains largely unknown. In this work, we present the characterization of *Citrobacter* sp. T1.2D-1, isolated from a 63.6 m-deep core sample extracted from the deep subsurface of the Iberian Pyrite Belt (IPB). A genomic analysis was performed to identify genes that could be ecologically significant in the IPB. We identified all the genes that encoded the formate–hydrogen lyase and hydrogenase-2 complexes, related to hydrogen production, as well as those involved in glycerol fermentation. This is particularly relevant as some of the substrates and byproducts of this process are of industrial interest. Additionally, we conducted a phylogenomic study, which led us to conclude that our isolate was classified within the *Citrobacter telavivensis* species. Experimentally, we verified the strain's ability to produce hydrogen from glucose and glycerol and, thus, of performing dark fermentation. Moreover, we assessed the activity of the nitrate and tetrathionate reductase complexes and the isolate's ability to tolerate high concentrations of heavy metals, especially Zn. These results suggest that *C. telavivensis* T1.2D-1 can play a role in the carbon, hydrogen, iron, nitrogen, and sulfur cycles that occur in the deep subsurface of the IPB, making it a candidate worthy of further study for possible biotechnological applications.

**Keywords:** *Citrobacter*; deep subsurface; Iberian pyrite belt; biohydrogen; dark fermentation



## 1. Introduction

The production of hydrogen via dark fermentation encompasses processes performed by microorganisms under aphotic and anaerobic conditions using organic substrates. The interest in this approach has increased due to the possibility of using waste biomass or wastewater as substrates, which would not only reduce hydrogen's production costs, but can also offer a sustainable way to generate clean energy [1]. However, the importance of hydrogen is not limited to its potential as a renewable energy source; there are multiple environments where this gas is essential to microbial communities' well-being, including hot springs, the gut microbiome, soils, groundwaters, or the deep subsurface, among others [2–6]. The deep subsurface, in particular, is characterized by the absence of light and oxygen, as well as higher temperatures and pressure as the depth increases [6]. In this environment, the mineralogical composition and hydrology greatly influence life since the lack of organic matter becomes a constraint, and minerals may be the only substrates available for microbial development [7]. These conditions collectively render the deep subsurface an extreme environment, often referred to as the dark biosphere, where microorganisms develop completely isolated from the surface [8].

Subsurface ecosystems that thrive in the absence of photosynthesis-derived products are known as Subsurface Lithoautotrophic Microbial Ecosystems (SLiMEs). At present, the most accepted model holds that the primary producers, which are part of the dark

biosphere, are powered by hydrogen [9,10]. Moreover, the crucial role that this gas plays in underground communities was reported [5,9,11,12], and the existence of hydrogen produced by microorganisms within these habitats was also demonstrated [13]. This is the case, for instance, of the Iberian Pyrite Belt (IPB), one of the world's largest sulfide deposits. Located in the southeast of the Iberian Peninsula, between Huelva (Spain) and Grándola (Portugal), it is 240 km long by 35 km wide and Rio Tinto rises from its core in Peña de Hierro (Huelva, Spain) [14].

Río Tinto is a river renowned for its high content of heavy metals, mainly ferric iron, acidic pH (with a mean of 2.3), and rich microbial diversity [15]. For many years, these extreme conditions were attributed to mining activity, which has been ongoing in the area for over 5000 years [11]. Nevertheless, there are indications that these characteristics predate human intervention. Recent studies have suggested that the IPB hosts an underground bioreactor. The discharge from this bioreactor contains microbial metabolic byproducts that are believed to be responsible for the extreme conditions observed in the river's basin [16]. In particular, these characteristics have been associated with a reaction known as Nitrate-Dependent $Fe^{2+}$ Oxidation (NDFO) [11]. It is still unclear whether enzymes are involved or if it is an abiotic process governed by nitrogen reactive forms [17]. In either case, it is believed that the ferric iron generated in this reaction participates in the dissolution of pyrite under anaerobic conditions, producing protons, sulphate, and ferrous iron, which is oxidized by the NDFO creating a cycle [18–20]. This can explain the high concentration of these compounds in Río Tinto, as well as its acidic pH. In order to investigate the geological processes occurring in this ecosystem, the Iberian Pyrite Belt Subsurface Life Detection (IPBSL) drilling project was established. Two boreholes, BH10 and BH11, were drilled and sampled, reaching depths of 620 and 340 m, respectively. As a result of this study, the vast microbial diversity that hosts the subsurface of the IPB was assessed, and several microorganisms were isolated under anaerobic conditions. Some of them have already been characterized through either in vitro or in silico approaches and their adequacy in participating in various biogeochemical cycles within this ecosystem demonstrated [11].

The strain labeledT1.2D-1 was isolated from 63.6 m-deep samples from borehole BH11 using culture media designed for the enrichment of chemolithoautotrophic denitrifying microorganisms and an atmosphere consisting of $H_2$:$CO_2$. The preliminary analysis indicated that the strain was a member of the *Citrobacter* genus [21], which consisted of rod-shaped, Gram-negative, facultative anaerobic bacteria classified within the family *Enterobacteriaceae*. Traditionally, these microorganisms are studied for their clinical relevance [22]. However, they have also been isolated from different environmental samples, including sediments [23,24], water [25], and wastewater [26]. More recently, this genus has been further explored for its potential applications in biotechnology and the industry [27,28].

In this study, our focus is to explore the relevant features that can explain the ecological role of *Citrobacter* sp. T1.2D-1 in the deep subsurface of the IPB, with a particular emphasis on its fermentation potential. To achieve this, we conduct a genomic analysis that allows us to identify genes of ecological or biotechnological significance. The metabolic activities of interest are subsequently verified experimentally, proving the ability of the strain T1.2D-1 to perform dark fermentation using various substrates and to participate in the different biogeochemical cycles occurring in the subsurface of the IPB.

## 2. Materials and Methods

### 2.1. Methodology for Citrobacter sp. T1.2D-1 Isolation

The methodology followed for drilling borehole BH11 is explained by Amils et al. (2013) [29]. The isolation of the strain T1.2D-1 was performed using ~6 g of powdered rock from a 63.6 m-deep core sample extracted from borehole BH11 as inoculum. The isolation procedure was performed in an anaerobic chamber using anaerobic culture media for the enrichment of chemolithoautotrophic denitrifying microorganisms composed of 0.3 g/L $NH_4Cl$, 0.3 g/L $K_2HPO_4 \cdot 3H_2O$, 0.1 g/L $MgSO_4 \cdot 7H_2O$, 0.4 g/L $NaHCO_3$, 0.01 g/L $CaCl_2 \cdot 2H_2O$, 0.1 g/L yeast extract, 40 mM nitrate, 20 mM thiosulfate, and 1 ml/L of trace

element solution [30] at an initial pH of 7 and an atmosphere composed of $H_2$:$CO_2$ (80:20, *v/v*). The isolation procedure is described by Leandro et al. (2018) [31].

### 2.2. DNA Extraction and Sequencing

Genomic DNA was extracted from an aerobic culture of the strain T1.2D-1 grown in Tryptic Soy Broth (TSB) using the Cetyltrimethylammonium Bromide (CTAB)-based method [32]. The final DNA concentration was determined with a Qubit v.2.0 fluorometer (Thermo Fisher, Scientific ES, Madrid, Spain). Library preparation and sample sequencing were performed by MicrobesNG (Birmingham, UK) using the Illumina MiSeq platform.

### 2.3. Genomic Analysis

The quality of paired-end reads was assessed using the Comprehensive Genome Analysis tool of the BV-BRC database v3.29.20 [33]. Genome assembly and annotation were conducted using Galaxy v23.0.rc1 [34]. In particular, SPAdes v3.15.4 was used to perform de novo assembly in contigs of the sequences previously trimmed by MicrobesNG, and plasmidSPAdes v3.15.4 to search for and assemble putative plasmids [35,36]. For contigs generation, all the parameters were maintained in their default values except k-mers, using 49, 91, 101, 121, and 127. Contigs were merged into scaffolds using SSPACE v3.0 software for Linux [37]. The assembled genome was represented using the Circular Viewer tool of BV-BRC.

Genome annotation was conducted using the RAST platform v2.0 [38], keeping all the parameters in their default values. Gene classification into different subsystems was visualized using the Seed viewer tool. The search for proteins related to metabolisms of interest was conducted using the BLAST protein tool from the NCBI server. The sequences of these proteins were retrieved from the UniProt database. We considered a positive result whenever the query cover was higher than 50%, the E-value less than $10^{-5}$, and the percent identity higher than 35%.

A phylogenomic analysis was performed through the calculation of Digital DNA-DNA Hybridization (dDDH), Average Nucleotide Identity (ANI), and Average Amino acid Identity (AAI). The dDDH values were estimated using the Genome-to-Genome Distance Calculator (GGDC) from the Leibniz Institute DMSZ [39]. ANI and AAI values were determined with the NCBI Prok tool from the Microbial Genome Atlas (MiGA) [40]. The phylogenomic tree was built with the BV-BRC Bacterial Genome Tree tool, using 1000 genes, 100 bootstrap replicas, and keeping the rest of the parameters as default values. The genomes of the strains used were retrieved from the NCBI database. Archaeopteryx.js v2.0.0a6 was used to visualize the tree [41].

### 2.4. Utilization of Different Electron Donors under Aerobic and Anaerobic Conditions

The electron donors used by *Citrobacter* sp. T1.2D-1 under aerobic and anaerobic conditions were determined in M9 medium, composed of 1 g/L $NH_4Cl$, 11 g/L $Na_2HPO_4 \cdot 7H_2O$, 3 g/L $KH_2PO_4$, 5 g/L NaCl, 4 g/L carbon source, 0.120 g/L $MgSO_4$, and 10 mg/L $CaCl_2$ [42]. The pH was adjusted to 7, before autoclaving with NaOH 1M. The carbon sources used under aerobic conditions were acetate, arabinose, citrate, fructose, galactose, glucose, glycerol, lactate, mannitol, pyruvate, starch, and sucrose; under anaerobic conditions, they were acetate, citrate, galactose, glucose, glycerol, lactose, malic acid, pyruvate, and sucrose. A total of 1 ml of a culture in exponential growth in medium PaFe2N2 [43] supplemented with 10 mM of acetate and 5 mM of nitrate for the anaerobic assays, or in medium TSB for the aerobic assays, were inoculated into 15 ml of medium M9. The experiments were conducted at room temperature in duplicates plus a non-inoculated negative control. The aerobic cultures were incubated in a shaker at 130 rpm whilst the anaerobic ones were left static.

Growth was determined by measuring the optical density at 600 nm using a Thermo Scientific Evolution 300 UV-Visible spectrophotometer. Durham tubes were used for the detection of gas production.

### 2.5. Evaluation of the Consumption and Production of Hydrogen

The consumption of hydrogen by *Citrobacter* sp. T1.2D.1 was tested in M9 medium using fumarate as the carbon source and an atmosphere composed of $H_2$:$N_2$ (80:20, *v/v*). Its production was evaluated in the same medium; however, glycerol and glucose were added as electron donors in an atmosphere of $N_2$. The production experiments were inoculated with 1 ml of the strain in exponential growth in medium M9 supplemented with glucose or glycerol into a volume of 15 ml. The consumption experiment was inoculated with a culture grown in medium PaFe2N2 using the same volumes as the production assay. Both experiments were performed at room temperature and in triplicates. Additionally, the glucose concentration was increased to 10 g/L and glycerol to 40 g/L. The production of gases was detected with a Bruker Series Bypass 450GC chromatograph. It was equipped with a column CP2056 0.6 m x 1/8'Ultimetal Cromsorb GHP 100–120 mesh and a column CP81073 0.5 m × 1/8' Ultimetal Hayesep Q 80–100 mesh; a detector TCD at 200 °C for the detection of $H_2$ and $CO_2$ and a detector FID at 250 °C for the detection of $CH_4$. $N_2$ were used as carrier gases.

### 2.6. Nitrate-Dependent Fe$^{2+}$ Oxidation Assay

The NDFO assay was performed using PaFe2N2 medium with 5 mM of nitrate, 10 mM of acetate, and 4 mM of $FeCl_2$. The preparation of the medium, its inoculation, and measurements of iron oxidation were performed as explained by Mateos et al. (2022) [43]. The inoculated bottles were manipulated inside an anaerobic chamber in a nitrogen atmosphere (Coy Laboratory products, EEUU). Nitrate reduction was measured through a N-(1-Naphthyl)ethylenediamine (NEDA)-based method [44].

### 2.7. Tetrathionate Reduction Assay

To assess the ability of *Citrobacter* sp. T1.2D-1 to reduce tetrathionate, we used a medium composed of 13.6 g/L $KH_2PO_4$, 2 g/L $(NH_4)_2SO_4$, 0.2 g/L $MgSO_4 \cdot 7H_2O$, 2 g/L galactose, 0.3 g/L yeast extract, 0.3 g/L casamino acids, 1 g/L $K_2S_4O_6$, and 5 g/L KOH [45]. All the components were sterilized by autoclaving, except for tetrathionate, which was sterilized using syringe filters with a pore size of 0.22 μm (Nuclepore, Whatman, Maidstone, UK). A total of 1 ml of a culture in exponential growth cultured in PaFe2N2 supplemented with 5 mM of nitrate and 10 mM of acetate under anaerobic conditions was used as inoculum placed into 15 ml of the medium. A total of 1 ml of each culture was extracted for the measurement of tetrathionate and centrifuged at 4000 x g for 5 minutes. The supernatant was preserved at –20 °C until measurement. The reduction of tetrathionate was quantified using a modified version of the method proposed by Kelly et al. (1969) [46]. Briefly, 200 μL of sample were mixed with 320 μL of $NaH_2PO_4$ 0.2 M-NaOH 0.2 N buffer at pH 7.4, 280 μL of water, and 400 μL of 0.05 M KCN at 4 °C. The mixtures were incubated for 20 minutes at 4 °C and tempered for 15 minutes. Finally, 240 μL of 1.5 M $Fe(NO_3)_3 \cdot 9H_2O$ 0.5 M in 4 N $HClO_4$ and 560 μL of water were added. Spectrophotometric measurements were taken at 460 nm.

### 2.8. pH Tolerance Assay

The tolerance of *Citrobacter* sp. T1.2D-1 to different pH values, ranging from 3 to 10, was evaluated in duplicates using Luria Bertani (LB) broth as the basal medium at room temperature and a 130 rpm shaking speed. A total of 1 ml of an aerobic culture in exponential growth in medium TSB was used as inoculum placed into 15 ml of LB at each pH. pH adjustment was conducted after autoclaving using 1 M NaOH and HCl stocks sterilized by filtration and a Thermo Scientific Orion 920a pH meter. Growth was determined by measuring the optical density at 600 nm.

### 2.9. Heavy Metal-Resistance Assay

The heavy metal-resistance assay was performed in triplicate with a negative control in Tryptic Soy Agar (TSA) medium. After autoclaving and before gelation, the TSA medium

was supplemented with different concentrations of each metal using 0.5 M stocks of the different salts sterilized by filtration. The salts used were $CrCl_3 \cdot 6H_2O$, $CuSO_4 \cdot 5H_2O$, $CdSO_4 \cdot 8/3H_2O$, $NiCl_2 \cdot 6H_2O$, $CoSO_4 \cdot 7H_2O$, $Pb(NO_3)_2$, $ZnCl_2$, and $HgCl_2$ at concentrations of 0.1 mM, 0.5 mM, 1 mM, 5 mM, 7.5 mM, 10 mM, 12.5 mM, 15 mM, 17.5 mM, 20 mM, and 22.5 mM, respectively. The agar plates were incubated at room temperature for two weeks.

## 3. Results and Discussion

### 3.1. Genomic Analysis

#### 3.1.1. Genome Assembly and Annotation

We sequenced the genome of *Citrobacter* sp. T1.2D-1 to characterize it and identify the metabolic activities of interest. The sequencing run generated 951,331 paired-end reads of $2 \times 250$ bp resulting in a mean coverage of 85X. The bacterial chromosome was assembled into 81 contigs and 67 scaffolds, with a length of 5,435,771 bp and a GC content of 53.56% (Figure 1). No putative plasmids were identified. These results align with the known characteristics of the *Citrobacter* genus, which typically exhibits a genome size and GC content ranging from 4 to 5 Mb and 51–56%, respectively [47].

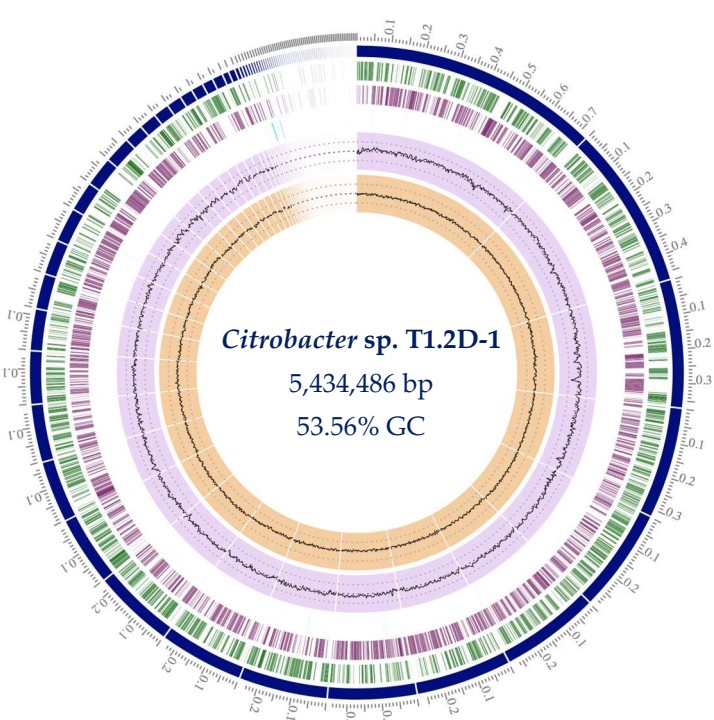

**Figure 1.** Circular representation of the genome of *Citrobacter* sp. T1.2D-1. From the outside to the inside, the following are shown: (1) the size of each contig in Mbp; (2) contigs in which the genome is assembled; (3) coding regions in the forward strand; (4) coding regions in the reverse strand; (5) non-coding regions; (6) GC content; (7) GC skew.

Genome annotation using the RAST platform found 80 tRNA genes, 12 rRNA genes, and 5,312 Coding DNA Sequences (CDSs). Nearly 50% of the genes were classified into the categories of carbohydrate metabolism, protein metabolism, and amino acid metabolism. To further investigate the metabolic diversity of strain T1.2D-1 and its ecological role in the subsurface of the IPB, we focused our study on identifying genes encoding enzymes that enabled the isolate to thrive under the extreme oligotrophic conditions in this ecosystem [11] (Table S1).

Starting with the hydrogen cycle, we found the presence of genes that encoded both the formate–hydrogen lyase (FHL) and hydrogenase-2 (Hyd-2) complexes. This finding positions *Citrobacter* sp. T1.2D-1 as a significant asset in the context of the subsurface of the IPB, as it has the ability to produce and consume hydrogen (Figure 2). The FHL

system consists of a formate dehydrogenase that transfers electrons to a [NiFe] hydrogenase encoded by the *hyc* genes. This complex has been extensively characterized and is known to be responsible for the hydrogen produced by some facultative anaerobes during mixed-acid fermentation [48]. On the contrary, the Hyd-2 complex can oxidize hydrogen in the periplasm coupled to the reduction of menaquinones, upon which the complex Fumarate Reductase (FRD) is dependent. Furthermore, the reversibility of this system has been reported in another strain of the genus [49], allowing hydrogen production during glycerol fermentation when the quinone pool is excessively reduced [50]. Therefore, *Citrobacter* sp. T1.2D-1 exhibits the potential to generate $H_2$ through fermentation using two different mechanisms depending on the carbon source.

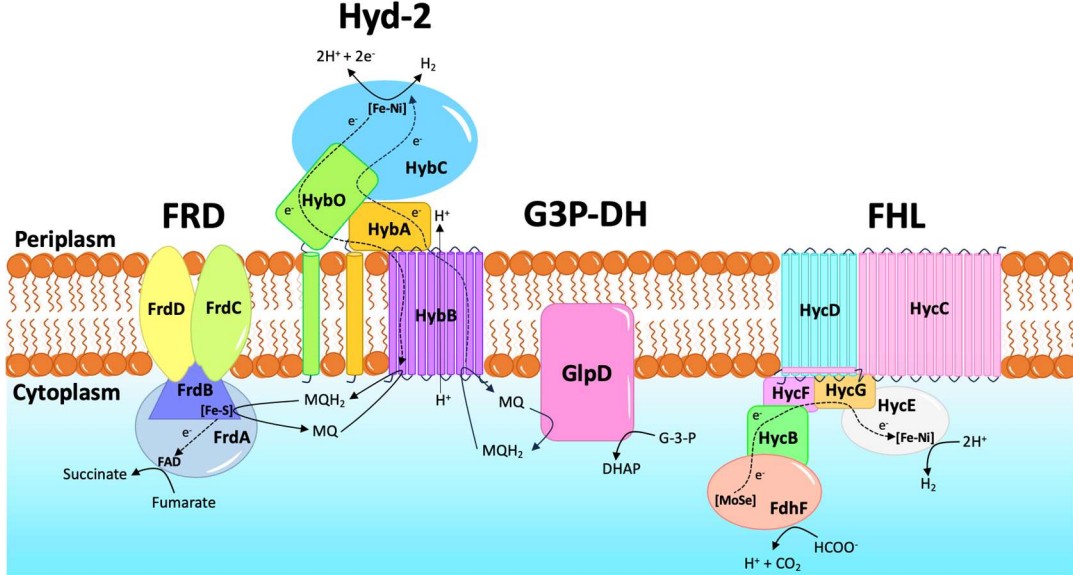

**Figure 2.** Complexes identified in the genomic analysis, which are involved in the production and consumption of hydrogen in *Citrobacter* sp. T1.2D-1. The consumption of hydrogen is represented as an interplay between hydrogenase-2 and fumarate reductase complexes. Hydrogen production is represented as a result of the activity of the formate–hydrogen lyase complex and of glycerol-3-phosphate dehydrogenase. Subunits of the enzymes are indicated. Electron flux is indicated with dashed arrows. Abbreviations: DHAP, dihydroxyacetone phosphate; FHL, formate–hydrogen lyase; FRD, fumarate reductase; G-3-P, glycerol-3-phosphate; G3P-DH, glycerol-3-phosphate dehydrogenase; Hyd-2, hydrogenase-2.

We also identified several activities that would allow the isolate to utilize different carbon sources under anaerobic conditions. The strain T1.2D-1 has the complete mixed-acid fermentation pathway, suggesting that it has the potential to ferment acetate, formate, oxaloacetate, fumarate, malate, or citrate. We also identified the *frc* gene, involved in the conversion of oxalate to formate, both substrates found in the IPB [11]. The production of $CO_2$ via this pathway holds ecological significance since it can be used in the production of organic matter, acetate, or methane by microorganisms capable of performing carbon fixation, acetogenesis, or methanogenesis, respectively, contributing to the survival of fermenters or methanotrophs. Likewise, we delved into the ability of the isolate to ferment glycerol, as it is a process that is arousing increasing interest in the industry [51]. *Citrobacter* sp. T1.2D-1 has all the genes to convert glycerol into 1,3-propanediol (1,3-PDO) via a reductive pathway, which involves glycerol dehydratase and 1,3-PDO oxidoreductase activities, or into hydrogen through an oxidative route that implies the production of glycolytic byproducts and the mixed-acid fermentation route [52]. Therefore, strain T1.2D-1 can produce hydrogen when glycerol is used as a substrate via Hyd-2 or FHL complexes. Furthermore, 1,3-PDO is relevant since it can be used in the synthesis of biopolymers

from petrochemical compounds and in different industries, such as cosmetics or food [53]. Moreover, it has been reported that its production by microorganisms is advantageous in comparison to chemical synthesis [54]. In addition, we also identified the genes involved in the conversion of dihydroxyacetone phosphate into methylglyoxal and the posterior synthesis of 1,2-PDO, which is also highly demanded in the industry due to its wide range of applications [55].

Concerning the nitrogen cycle, we found the complete operon of nitrate reductase, indicating that the isolate had the ability to reduce nitrate to nitrite (Figure 3). Nitrite is particularly significant in the subsurface of the IPB as it has been associated with NDFO, which, as mentioned above, is considered one of the main factors contributing to the extreme conditions observed in the Río Tinto basin. Moreover, nitrate has been found in the subsurface of the IPB, as well as nitrite [11], meaning that the reaction would be feasible. Furthermore, the ability of strain T1.2D-1 to perform this reaction involves it in the iron cycle of the IPB.

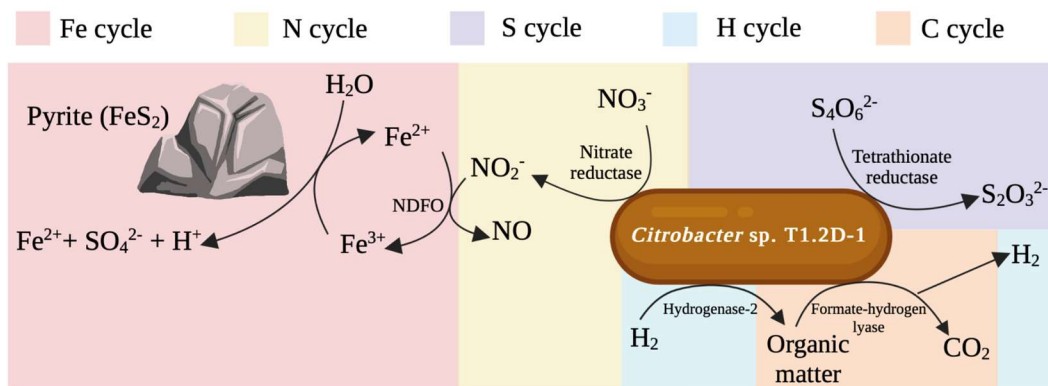

**Figure 3.** Overview of the biogeochemical cycles in which *Citrobacter* sp. T1.2D-1 can participate in the subsurface of the IPB based on the results of the genomic analysis. The name of the enzymatic activities involved is indicated in every reaction. Created with Biorender.

Lastly, regarding the sulfur cycle, which is of special relevance in this context since the IPB is one of the largest metal sulfide deposits on Earth, we found the complete tetrathionate reductase operon (Figure 3). These preliminary results prove that *Citrobacter* sp. T1.2D-1 can be metabolically active in the deep subsurface of the IPB, contributing to the sustenance of hydrogen, carbon, nitrogen, iron, and sulfur biogeochemical cycles identified in this ecosystem.

### 3.1.2. Phylogenomic Analysis

As mentioned above, a previous analysis using the 16S rRNA gene sequence indicated that the strain T1.2D-1 was classified within the *Citrobacter* genus. To further identify it taxonomically, we conducted a genomic-based taxonomic analysis through the calculation of phylogenomic indices, such as ANI, AAI, and dDDH. This procedure was performed against the NCBI database. The results show that our isolate belongs to the species *C. telavivensis*, owing to the fact that we obtained the highest values of ANI (99.92%), AAI (95%), and dDDH (99.8%) for the strain *C. telavivensis* 6105 compared to the rest of species of the genus (Table S2), and that the three parameters exceed the thresholds for species demarcation (ANI $\geq$ 95%; AAI $\geq$ 95–96%; dDDH $\geq$ 70%) [38,56,57]. To further confirm this result, we built a phylogenomic tree using the codon tree method, where *C. telavivensis* appeared again as the species immediately closest to the isolate (Figure S1).

### 3.2. Carbon Sources Utilization

Regarding the preliminary results for the carbon cycle, we studied the ability of *Citrobacter* sp. T1.2D-1 to use various carbon sources under aerobic and anaerobic conditions.

The results show that the isolate is able to use the majority of the carbon sources tested for both conditions (Table 1), providing evidence that several reactions of the mixed-acid fermentation route are active, and that the strain T1.2D-1 can potentially use diverse carbon sources detected in the subsurface of the IPB, such as formate or oxalate.

**Table 1.** Use of different electron donors by *Citrobacter* sp. T1.2D-1 under aerobic or anaerobic conditions. Carbon sources in which we detected the proliferation of the strain are marked with "+" and those in which there is no growth with "−". The absence of data is indicated as "ND" and the electron donors in which we detected gas as fermentation product are marked with (G).

| Carbon Source | Growth | |
|---|---|---|
| | **Aerobic Conditions** | **Anaerobic Conditions** |
| Acetate | + | + |
| Arabinose | + | ND |
| Citrate | + | − |
| Fructose | + | ND |
| Fumarate | ND | + |
| Galactose | + | + (G) |
| Glucose | + | + (G) |
| Glycerol | + | + (G) |
| Lactate | + | ND |
| Lactose | ND | + |
| Malic acid | ND | + |
| Mannitol | + | ND |
| Pyruvate | + | + |
| Starch | + | ND |
| Sucrose | + | + (G) |

These results confirm that *Citrobacter* sp. T1.2D-1, in accordance with the characteristics of its genus [22], can ferment a wide range of substrates, including low-value substrates, such as glycerol. At present, high quantities of glycerol are generated as a byproduct of other industrial processes. Consequently, this surplus needs to be disposed of or transformed into products with a higher value, which can be performed through fermentation [58]. The fermentation of glycerol, as previously mentioned, can produce 1,3-PDO and 1,2-PDO, which have a higher industrial value. Therefore, it would be interesting to study the ability of the isolate to produce these two compounds. Furthermore, we can also study if the isolate can also ferment other low-value substrates, such as wastewater, waste biomass, or agricultural biomass, which have already been described as fermentation substrates [59–61].

The fermentative ability of *Citrobacter* sp. T1.2D-1 is also interesting for its products. For instance, since it can perform mixed acid fermentation, *Citrobacter* sp. T1.2D-1 can produce compounds with industrial interest, such as lactate, used both in the food and non-food industries [62], or ethanol, which is has gained more interest in recent years as it can be used as a biofuel [63].

Hydrogen Production

In the previous section, we detected, through Durham tubes, the production of gases in certain fermentations. To identify the gas and quantify its production, we performed gas chromatography using anaerobic cultures, where glucose and glycerol were used as sole carbon sources. The *Citrobacter* genus is known to be one of the most studied and promising hydrogen producers due to its ability to produce this gas from a wide range of substrates and various environmental conditions [64,65]. We observed that strain T1.2D-1 produced hydrogen at room temperature at an initial pH of 7 using glucose and glycerol as electron donors (Figure 4).

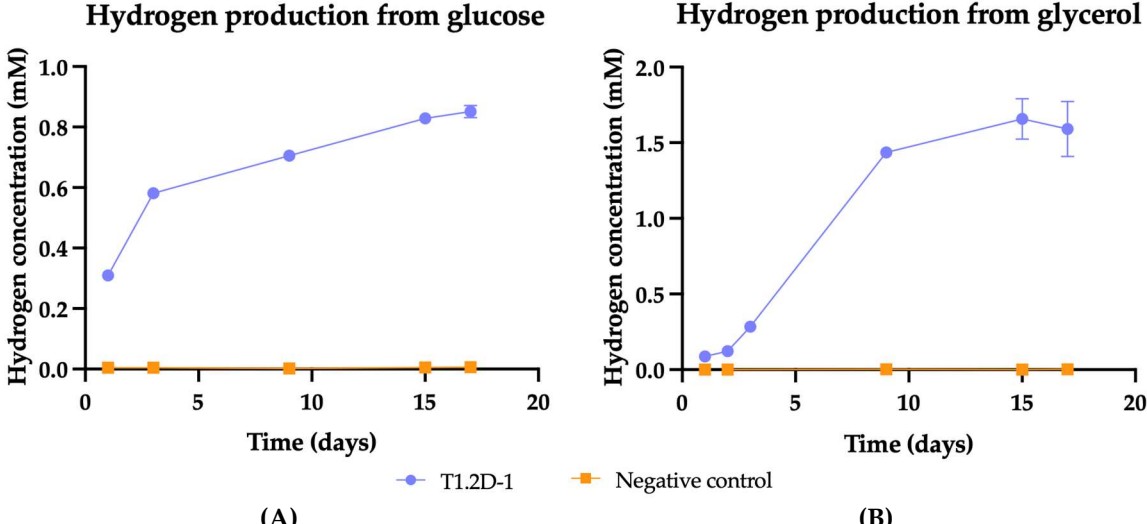

**Figure 4.** Hydrogen production from glucose (**A**) and glycerol (**B**) by *Citrobacter* sp. T1.2D-1. The variation in hydrogen concentration in the inoculated replicas (blue) is represented as a function of the negative control (orange). The standard deviation is indicated with error bars.

Hydrogen is of great significance in the deep subsurface because the absence of light impedes the development of photosynthetic microorganisms capable of generating organic matter for heterotrophs to consume; thus, alternative energy sources are needed. Thus, it might be valuable to examine this ability further and evaluate if, in similar conditions as those found in the subsurface, *Citrobacter* sp. T1.2D-1 can also produce this gas, since in IPB, sugars, such as glucose, mannose, and sorbitol, have been detected (V. Parro, personal communication) and, as mentioned above, biohydrogen production has been detected in deep subsurface samples [11,13]. Furthermore, it would be interesting to perform an optimization of the conditions of hydrogen production and calculate the maximum yield of the isolate, in order to compare it with the ones described for other strains of the genus and to explore the possible applications.

### *3.3. Other Metabolisms of Interest*

The genomic analysis showed that the strain T1.2D-1 could be metabolically active in the IPB. This is supported by the identification of multiple genes known to be involved in the biogeochemical cycles that occur in this system [11]. With the aim of testing the metabolic activities identified in the genome and how *Citrobacter* sp. T1.2D-1 can thrive in an extreme environment, such as the deep subsurface, several assays were designed and performed.

### 3.3.1. Nitrate-Dependent $Fe^{2+}$ Oxidation Assay

We aimed to confirm the activity of the nitrate reductase operon and the ability of the strain to perform the NDFO. A medium supplemented with nitrate as the electron acceptor, acetate as the electron donor, and $FeCl_2$ was used. We observed that, in a two-week period, *Citrobacter* sp. T1.2D-1 reduced nearly half of the nitrate, producing nitrite. Concurrently, we confirmed the oxidation of ferrous iron (Figure 5).

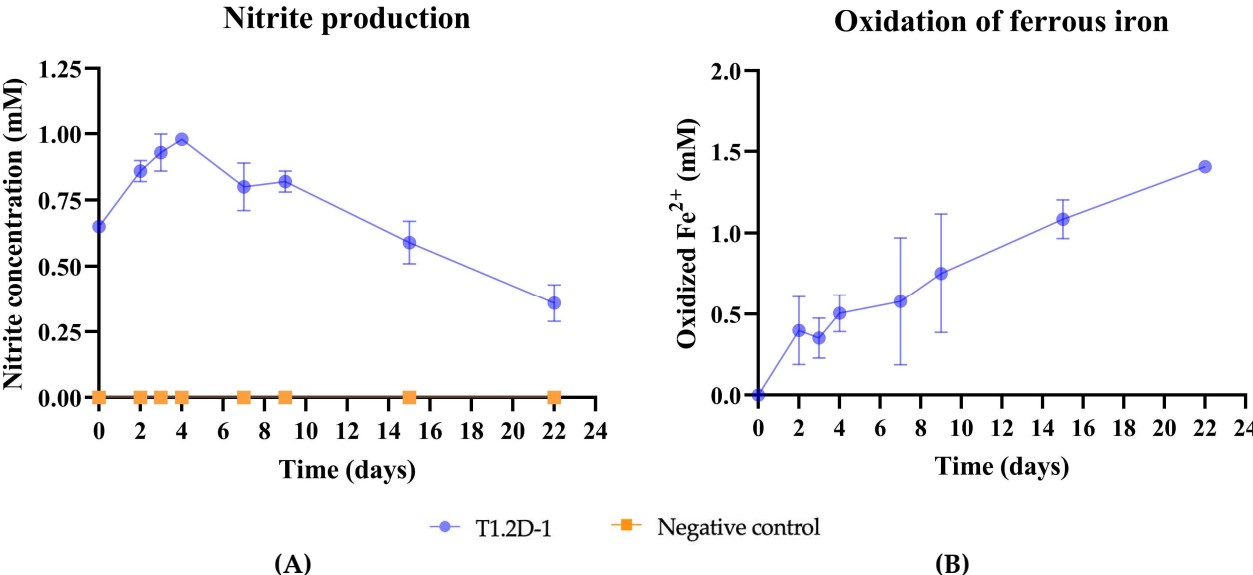

**Figure 5.** Results of the NDFO assay. (**A**) Graphic representation of the production of nitrite in the inoculated replicas (blue) versus the negative control (orange); (**B**) graphic representation of the $Fe^{2+}$ oxidized throughout the experiment in the inoculated replicas. In both graphs, the error bars represent the standard deviation.

This is the second time that it has been reported that a strain of the *Citrobacter* genus is capable of performing the NDFO [66]. These results emphasize that the deep subsurface of the IPB harbors microorganisms capable of performing reactions associated with the extreme conditions detected in the Río Tinto basin and iron cycling, as explained in Section 3.1. To explore this matter more thoroughly, it would be valuable to investigate how *Citrobacter* sp. T1.2D-1 interacts with pyrite. Studying whether the isolate can participate in its dissolution [18–20], as well as its ability to interact with the mineral surface [67], would help to gain a further insight into the ecology of the deep subsurface of the IPB.

3.3.2. Tetrathionate Reduction Assay

As previously mentioned, in the genomic analysis, we identified the complete tetrathionate reductase operon. In order to prove the activity of this enzyme, we cultured *Citrobacter* sp. T1.2D-1 in a medium supplemented with galactose as the electron donor and tetrathionate as the electron acceptor under anaerobic conditions. We measured the concentration of tetrathionate in the medium and observed that the strain was capable of effectively reducing all of the tetrathionate present within a two-day period (Figure S3). This makes *Citrobacter* sp. T1.2D-1 a likely candidate for the sulfur cycle of the IPB. The production of thiosulfate through these mechanisms holds ecological relevance since this element can be utilized in the production of sulfates or sulfides. Notably, both elements have already been detected at various depths within the IPB and can serve as substrates for some of the microorganisms identified in this ecosystem [11].

3.3.3. Hydrogen Consumption Assay

With the aim of further investigating the strain's metabolic diversity, we studied whether *Citrobacter* T1.2D-1 had the ability to use hydrogen in the presence of fumarate. The isolate was cultured in medium supplemented with fumarate and an $H_2:N_2$ atmosphere. We observed that, within two weeks, the strain used nearly half of the hydrogen present in the headspace of the serum bottles (Figure 6), proving the activity of the Hyd-2 complex and the ability of the isolate to use this gas as an electron donor.

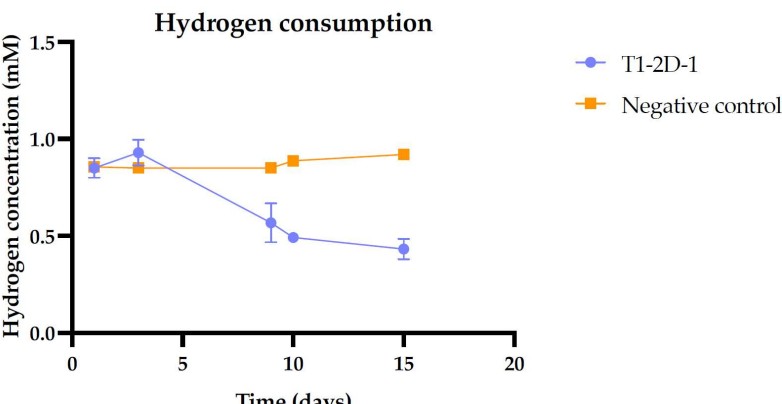

**Figure 6.** Hydrogen consumption by *Citrobacter* sp. T1.2D-1. The variation in hydrogen concentration in the inoculated replicas is represented as a function of the negative control. The typical deviation for each measure is indicated with error bars.

Notably, recent articles have reported *Citrobacter* strains capable of performing carbon fixation under heterotrophic conditions [68,69]. This process is associated with the presence of a homolog of the *rnfC* gene of *Acetobacterium woodii*, which is also identified in the genome of *Citrobacter* sp. T1.2D-1, and with the ability to use hydrogen as an electron donor via Hyd-2 [69]. Therefore, it would be interesting to study whether strain T1.2D-1 is also capable of performing this reaction, since in an environment, such as the deep subsurface, this can be of great importance when carbon sources are scarce.

### 3.3.4. pH Tolerance Assay

Within the first 150 m of borehole BH11, the pH was determined as acid, ranging from 3.5 to 6.5 (R. Amils, personal communication). In particular, at a 63.6 m-depth from which *Citrobacter* sp. T1.2D-1 was isolated, the pH was nearly 4. We determined that the isolate was capable of growing at pH 4.0–9.0 in LB (Figure S2), and thus the strain T1.2D-1 was capable of tolerating the acidic conditions described for some of the depths of the BH11.

### 3.3.5. Heavy Metal Tolerance Assay

As stated previously, Río Tinto is known for the high concentration of heavy metals in its waters, elements that have also been reported in the subsurface of the IPB [11]. Additionally, members of the *Citrobacter* genus have been isolated from environments where these compounds are present and have been studied with bioremediation purposes [70]. For these reasons, we considered it an interesting proposition to test whether strain T1.2D-1 was capable of tolerating high concentrations of heavy metals. We determined that the isolate could grow in medium supplemented with 1 mM of $Hg^{2+}$, 7.5 mM of $Co^{2+}$ and $Cu^{2+}$, 10 mM of $Ni^{2+}$, 12.5 mM of $Cd^{2+}$, $Cr^{3+}$, and $Pb^{2+}$, and 20 mM of $Zn^{2+}$. The ability of *Citrobacter* sp. T1.2D-1 to resist high concentrations of heavy metals is clearly superior to those reported for other enterobacteria, such as *Escherichia coli*-type strains DH5$\alpha$ and K-12 [71,72], as well as other strains of the *Citrobacter* genus [70]. In particular, we found a remarkable tolerance to $Ni^{2+}$ and $Zn^{2+}$. Only one *Citrobacter* strain, belonging to the *Citrobacter intermedius* species, was described as being able to grow in higher concentrations of $Ni^{2+}$ (with a minimal inhibitory concentration of 15 mM) [73], and none were reported with a tolerance to $Zn^{2+}$ as high as the one determined for T1.2D-1. Zn has been referred to as a soil contaminant of concern. Therefore, taking into account the tolerance of *Citrobacter* sp. T1.2D-1 to Zn, it might be interesting to look at this ability in more detail, as this isolate can also be potentially used in the bioremediation of Zn [74].

Moreover, this noteworthy tolerance also has a genetic basis. We found multiple genes associated with resistance to these types of compounds, such as *cusRS* and *cusCFBA,* the former being a response system that induces the expression of the latter, which functions as a pump that retires the excess of Cu (I) from the periplasmic space of *E. coli*. We also

identified the transcriptional regulator *cueR*, which enhances the expression of the also present *copA*, whose function is to transport Cu (I) from the cytosol to the periplasm, where it is reduced to Cu (II) (a less toxic form) by CueO [75]. We also identified *nikABCDE* operon, which encodes an Ni import system [76], and *rcnA*, which is translated into a membrane-associated protein whose presence has been related to an increase in resistance to Co and Ni. Lastly, focusing on the mechanisms of tolerance to Zn, we found *zntA* and *zntR*, which coded for a type-P ATPase that was predicted as the first Zn-specific transporter, responsible for the release of Zn ions out of the cell. In addition, ZntA could also be activated by Cd, Ag, and Pb, translocating this last element [77]. Conversely, we identified two Zn-uptake systems, another P-type ATPase and a high-affinity periplasmic-binding protein encoded by the genes *zosA* and *znuABC* [74,77], respectively, as well as *zupT*, which were not only involved in importing this metal into the cell but also of Cd, Fe, Co, and possibly Mn, given its broad specificity [78]. These results show that the strain T1.2D-1 is adapted to the presence of heavy metals, as expected from the environment in which it lives.

## 4. Conclusions

In this study, we reported the first *Citrobacter* strain isolated from hard-rock samples extracted from the deep subsurface. The genomic analysis indicates that strain T1.2D-1 belongs to the *Citrobacter telavivensis* species and helps identify the enzymatic activities of ecological significance. The experimental verification of features, such as hydrogen production and consumption, the fermentation of various sugars, nitrate reduction coupled to iron oxidation, and tetrathionate reduction, demonstrates that *Citrobacter telavivensis* T1.2D-1 can be metabolically active in the deep subsurface of the IPB, and that it may play a role in the hydrogen, carbon, nitrogen, iron, and sulfur biogeochemical cycles. Furthermore, the possibility of the isolate being metabolically active in this ecosystem is supported by its ability to tolerate high concentrations of heavy metals and a low pH, conditions characteristic of Río Tinto and the IPB. These results suggest that *Citrobacter telavivensis* T1.2D-1 is worthy of further study, as potential biotechnological applications for this strain can emerge. In particular, delving into some of its most remarkable features, such as its capacity to produce hydrogen and substrates of biotechnological interest, as well as its ability to withstand high concentrations of heavy metals, may very well reveal the potential uses of the strain in the industry and bioremediation, respectively.

The deep subsurface hosts, such enormous microbial diversity, remain, at present, uncharacterized. Since hydrogen production via dark fermentation becomes increasingly plausible for clean energy production, it is imperative to study different aspects with the aim of finding candidates that can take part in this process. In this respect, it is not unreasonable to think that some of them are members of the dark biosphere, giving us yet another reason to continue exploring the life beneath our feet.

**Supplementary Materials:** The following supporting information can be downloaded at: https://www.mdpi.com/article/10.3390/fermentation9100887/s1, Figure S1: phylogenomic tree built with the codon tree method using representative strains of the *Citrobacter* genus; Figure S2: growth of *Citrobacter* sp. T1.2D-1 at different pH values; Figure S3: tetrathionate reduction by *Citrobacter* sp. T1.2D-1; Table S1: genes related to metabolisms of interest identified in the genome of *Citrobacter* sp. T1.2D-1; Table S2: values of the phylogenomic analysis conducted comparing *Citrobacter* sp. T1.2D-1 with the closest isolates of the genus.

**Author Contributions:** Conceptualization, V.G.-R. and A.M.-B.; methodology, V.G.-R., A.M.-B. and N.R.; formal analysis, V.G.-R. and A.M.-B.; investigation, V.G.-R., A.M.-B. and N.R.; data curation, V.G.-R.; writing—original draft preparation, V.G.-R.; writing—review and editing, A.M.-B. and R.A.; supervision, R.A.; funding acquisition, R.A. All authors have read and agreed to the published version of the manuscript.

**Funding:** This research was funded by the Spanish Ministry of Science and Innovation (projects TED2021-129563B-I00 and PID2022-136607NB-I00). A.M.B. acknowledges the Grant FPU19/01743 from the Spanish Ministry of Universities.

**Institutional Review Board Statement:** Not applicable.

**Data Availability Statement:** Reads have been deposited at DDBJ/ENA/GenBank under the accession number ERR11267709, and the complete genome sequences and annotations have been deposited under the accession numbers CATNHH010000000 for this chromosome and GCA_950098555.1 for the annotation.

**Acknowledgments:** The authors would like to thank D. Ruano for providing valuable insights during the development of this work, T. Leandro for the isolation of the strain, V. Parro for the unpublished results, the IPBSL team for obtaining the data from the drilled borehole, and R. Samalot and N. Aldaba for improving the English of this manuscript.

**Conflicts of Interest:** The authors declare no conflict of interest.

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
