# Peer review of "Dark Fermentation in the Dark Biosphere: The Case of Citrobacter sp. T1.2D-12"

_fermentation, doi:10.3390/fermentation9100887_

Round 1
Reviewer 1 Report
I review the article on title “Dark fermentation in the dark biosphere: the case of 2 Citrobacter sp. T1.2D‐1” which needs revision before publication, see below comments:
1. Title needs revision “ what difference between ‘ Dark fermentation and dark biospare”
2. Revise the sentence line 11-12.Members of the Citrobacter genus have been isolated from a wide range of environments some of which are known to exhibit diverse metabolic activities of interest.
3. Revise the sentence line 31-33.However, the importance of hydrogen is not limited to industry; there are multiple environments where this gas is essential to sustain the microbial communities that inhabit them.
4. Revise the sentence line 462-465.
5. Line# 134 what is this? “medium PaFe2N2”
6. Please check all abbreviation and explain it first time
7. This articles have many grammatical mistakes, please revised it carefully.
8. Please used recent references revised reference numbers 9,10,15,18,20,29,37,45,48 and 56
Moderate editing of English language required
Author Response
ANSWERS TO THE REVIEWERS
In italics the reviewers comments
Rev 1
I review the article on title “Dark fermentation in the dark biosphere: the case of 2 Citrobacter sp. T1.2D‐1” which needs revision before publication, see below comments:
- Title needs revision “ what difference between ‘ Dark fermentation and dark biospare”
We have discussed more in detail the requested definitions in the introduction. We hope they are enough to properly understand both terms. We prefer to keep the original title but if the reviewer prefers to use deep subsurface biosphere we can change the title to “Dark fermentation in the deep subsurface biosphere: the case of Citrobacter sp.T1.2-D1”
- Revise the sentence line 11-12.Members of the Citrobacter genus have been isolated from a wide range of environments some of which are known to exhibit diverse metabolic activities of interest.
Since we have modified the abstract, we hope that this problem has been solved.
- Revise the sentence line 31-33.However, the importance of hydrogen is not limited to industry; there are multiple environments where this gas is essential to sustain the microbial communities that inhabit them.
We have modified the sentence to solve the aforementioned problem.
- Revise the sentence line 462-465.
The sentence has been revised as suggested.
- Line# 134 what is this? “medium PaFe2N2”
This is the name of a medium and not an abbreviation. We have moved the reference of the medium to that line, where it is mentioned for the first time.
- Please check all abbreviation and explain it first time
We have checked them and made the necessary changes.
- This articles have many grammatical mistakes, please revised it carefully.
Revised version has been checked for grammatical mistakes.
- Please used recent references revised reference numbers 9,10,15,18,20,29,37,45,48 and 56
We consider that these references are necessary as they give information about certain protocols or they are the original references in which information of relevance for the article was published for the first time.
9 Moderate editing of English language required
English language has been edited by a native English speaker.
Reviewer 2 Report
In this study, the genome of Citrobacter T1.2D-1 isolated from the subsurface of the Iberian Pyrite Belt (IPB) was characterized. The experiments were conducted under different conditions to evaluate the ecological role of Citrobacter T1.2D-1 in the deep subsurface of IPB. The results suggest that Citrobacter T1.2D-1 can be used for dark fermentation and can participate in different biogeochemical cycles in the deep subsurface of IPB, making it a candidate material for biotechnology applications. However, necessary revision is still required before its final publication.
Identification of the isolated species is a great issue. Was the identification method complete and reliable?
The abstract needs to be improved more accurately.
The introduction should be closely related to the title, including the isolation, the genomic analysis, the ecological role, and the possible applications of strains.
Some relavent studies on the topic of biohydrogen production can be cited, e.g. Journal of Environmental Management, 2022, 319, 115655; Chemosphere, 2022, 286(1), 131655; Bioresource Technology, 2021, 320(A): 124303.
The conclusions can be improved.
Author Response
Rev 2
In this study, the genome of Citrobacter T1.2D-1 isolated from the subsurface of the Iberian Pyrite Belt (IPB) was characterized. The experiments were conducted under different conditions to evaluate the ecological role of Citrobacter T1.2D-1 in the deep subsurface of IPB. The results suggest that Citrobacter T1.2D-1 can be used for dark fermentation and can participate in different biogeochemical cycles in the deep subsurface of IPB, making it a candidate material for biotechnology applications. However, necessary revision is still required before its final publication.
- Identification of the isolated species is a great issue. Was the identification method complete and reliable?
Yes, we sequenced the genome and used several phylogenomic indices to identify the strain. These indices are more reliable than the 16S identification as they employ the information contained in the whole genome.
- The abstract needs to be improved more accurately.
Abstract has been improved as suggested.
- The introduction should be closely related to the title, including the isolation, the genomic analysis, the ecological role, and the possible applications of strains.
The introduction has been changed to be closely related with the title and the reported research.
- Some relavent studies on the topic of biohydrogen production can be cited, e.g. Journal of Environmental Management, 2022, 319, 115655; Chermosphere, 2022, 286(1), 131655; Bioresource Technology, 2021, 320(A): 124303.
We have introduced one of the suggested references in the revised versión (Chemosphere 2022, 286(1), 131655.
- The conclusions can be improved.
Conclusions have been improved as suggested.
Reviewer 3 Report
The article "Dark fermentation in the dark biosphere: the case of Citrobacter sp. T1.2D‐1". The paper is in the interest of fermentation, however there are several issues to be solved before publication:
1. Line30, please explain what cost is reduced? Hydrogen production? Waste treatment? How did you lower it?
2. In Lines 31-33, the author does not indicate the importance of hydrogen in industry.
3. The author expressed the potential of Citrobacter sp. T1. 2D-1 in fermentation in the introduction, but there is too little discussion in the article. Please add more information in this section.
4. Improve the quality of the image and modify the X-axis scale for better observation.
5. The hydrogen concentration detection times in Figure 4a and b are inconsistent. Please explain the reason.
6. Based on the trend shown in Figure 4a, the hydrogen concentration was still increasing at day 17, and the line had not approached a plateau. Since the aim was to study the fermentation potential, the fermentation time should be extended according to the current trend until the hydrogen concentration approached stability. Please explain the reason for the experimental period setting.
7. Lines 354-355, insert references.
8. The experimental data in Figure 5b has high errors on days 7 and 9, and the validity of the experimental data is questionable.
9. Has the study considered the impact of geographic and environmental factors, as well as sampling location, on the metabolic activity and ecological role of Citrobacter sp. T1. 2D-1? If yes, how did these factors affect the research results?

Author Response
Rev 3
The article "Dark fermentation in the dark biosphere: the case of Citrobacter sp. T1.2D‐1". The paper is in the interest of fermentation, however there are several issues to be solved before publication:
- Line30, please explain what cost is reduced? Hydrogen production? Waste treatment? How did you lower it?
Cost reduction has been clarified.
- In Lines 31-33, the author does not indicate the importance of hydrogen in industry.
We have clarify this point in the revised version.
- The author expressed the potential of Citrobacter sp. T1. 2D-1 in fermentation in the introduction, but there is too little discussion in the article. Please add more information in this section.
Discussion on the potential of Citrobacter sp. T1.2-D1 has been expanded as suggested.
- Improve the quality of the image and modify the X-axis scale for better observation.
Image quality and modification of X-axis has been done as suggested.
- The hydrogen concentration detection times in Figure 4a and b are inconsistent. Please explain the reason.
The detection times were decided taking into account a previous experiment carried out in similar conditions. Although they are not identical, we think that they provide the intended information.
- Based on the trend shown in Figure 4a, the hydrogen concentration was still increasing at day 17, and the line had not approached a plateau. Since the aim was to study the fermentation potential, the fermentation time should be extended according to the current trend until the hydrogen concentration approached stability. Please explain the reason for the experimental period setting.
As mentioned before, based on a previous experiment we stopped taking measurements as the strain had reached at day 17 the maximum hydrogen production for that condition.
- Lines 354-355, insert references.
References have been introduced.
- The experimental data in Figure 5b has high errors on days 7 and 9, and the validity of the experimental data is questionable.
Despite the high error bars in those two days, we consider that this graph is necessary to demonstrate the ability of the isolate to oxidize iron.
- Has the study considered the impact of geographic and environmental factors, as well as sampling location, on the metabolic activity and ecological role of Citrobacter sp. T1. 2D-1? If yes, how did these factors affect the research results?
The analysis of the genome and the experiments carried out for this paper both had in mind the conditions found in the deep subsurface where the strain was isolated. Throughout the paper we discuss in several of the results with respect to the sampling location.
Round 2
Reviewer 2 Report
The authors have made good modifications on the manuscript. From my point of view, the revision can be accepted for the publication of Fermentation.
Personally suggest to carefully checking out the formatting issues, i.e. Line 422.